# ENDOASSISTANT: A LARGE-SCALE VISION- LANGUAGE DATASET FOR ENDOSCOPIC SURGERY UNDERSTANDING FROM OPEN-SOURCE VIDEOS

## ABSTRACT

Endoscopic interventions offer a minimally invasive approach, minimizing patient discomfort and facilitating expedited recovery. Proficient training of junior surgeons necessitates the ability to analyze and interpret endoscopic scenes through questioning and answering. Consequently, the development of a robust foundation model for endoscopic visual language understanding holds immense value for medical training and surgical education. However, existing endoscopy vision-language datasets are limited in scale and diversity, consisting of only 50 videos sourced from a few clinical sites, thus posing a significant hurdle to the advancement of generalized and robust artificial intelligence models for endoscopic surgical applications. To address this challenge, we present a large-scale, meticulously curated image-text dataset of surgical endoscopic scenes from expert surgeons, designed to propel a vision-language assistant in medical scene understanding. Encompassing 590 open-source videos spanning more than 91 hours, our curated dataset includes 65,844 unique images, 30,002 unique captions, and 157,589 image-caption/question-answering pairs. This dataset aims to assist the development of automated systems to support medical professionals by mitigating repetitive tasks. We present a comprehensive endoscopic surgery assisting pipeline, (1) a first-ever image-caption dataset specifically for endoscopic scenes; (2) an image-question-answer dataset that offers greater size and diversity compared to existing collections; (3) rigorous evaluation demonstrating its efficacy in downstream surgical endoscopic scene comprehension tasks like classification, retrieval and visual question answering.

## 1 INTRODUCTION

Endoscopic interventions (Sharata et al., 2013; Oblizajek et al., 2020; Bang et al., 2020) have revolutionized surgical practices by offering a minimally invasive alternative, which not only mitigates patient discomfort but also accelerates postoperative recovery periods. These procedures, facilitated by advancements in technology, entail the insertion of a slender, flexible tube equipped with a camera and light source into the body, allowing surgeons to visualize and operate within internal organs without the need for extensive incisions.

In the realm of surgical education, the proficiency of junior surgeons hinges on their ability to navigate and interpret the intricate scenes presented by endoscopic imagery. Imagine a novice surgeon, standing at the forefront of a surgical theater, peering into a monitor displaying a live feed from the endoscope inserted into a patient's body. They must swiftly discern anatomical structures, identify abnormalities, and make critical decisions in real time, all while under the pressure of time-sensitive procedures. This demands not only technical skill but also a deep understanding of the visual cues provided by the endoscopic images. Consequently, effective training programs are highly demanded to incorporate interactive learning approaches that simulate the dynamic nature of surgical settings. By engaging in iterative cycles of questioning and answering, junior surgeons can refine their ability to analyze endoscopic scenes, anticipate challenges, and formulate strategic responses. To this end, the development of a robust foundation model for endoscopy visual language understanding emerges as a transformative endeavor in surgical education. Such a model, leveraging

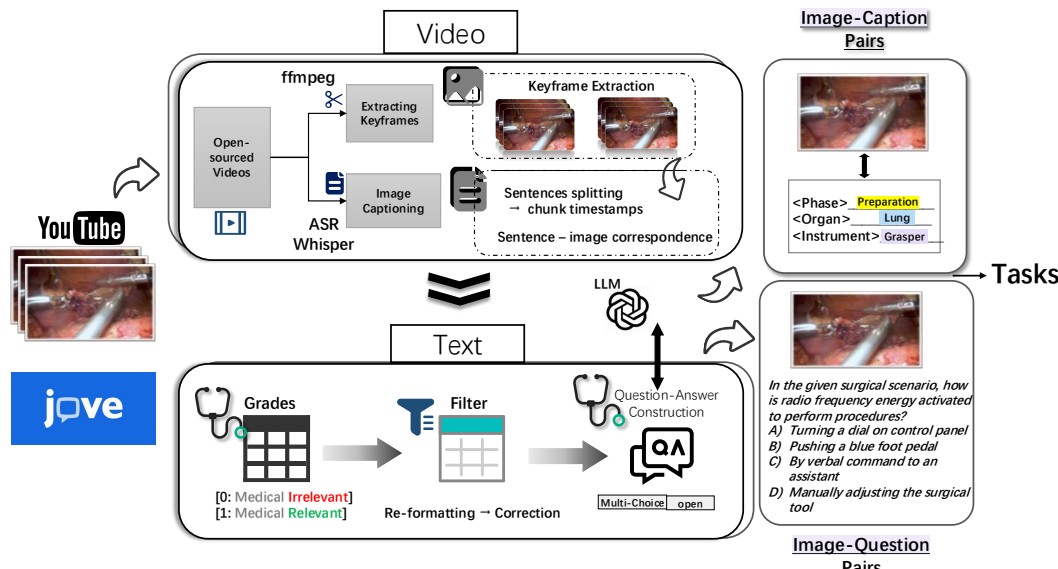

Figure 1: Working pipelines for data curation. We first collect large-scale open-sourced endoscopic surgical videos with captions from expert surgeons. The videos are chunked based on the extracted text sentences. Then keyframes are extracted and aligned with the text based on the timestamps. Both images and texts are graded and refined in collaboration with expert surgeons, supported by large vision and language models to ensure accuracy and consistency. Finally, image-caption pairs and image-question-answer pairs are generated for downstream applications including VQA, classification, and retrieval.

the power of artificial intelligence and machine learning, can systematically decode visual semantics embedded within endoscopic images.

Constructing a foundation model for endoscopic surgery understanding requires a comprehensive vision-language dataset for endoscopic surgery on a large scale. However, the current landscape of endoscopy vision-language datasets (*e.g.*, Cholec80-VQA (Twinanda et al., 2016), PSI-AVA-VQA (Valderrama et al., 2022b)) presents a significant limitation in both scale and diversity, typically comprising fewer than 50 videos from the single clinical site. This scarcity hinges on the progress toward the creation of generalized and robust artificial intelligence models tailored specifically for endoscopic surgical applications. To address the pressing need for comprehensive datasets in the field, we introduce **EndoAssistant**, a meticulously curated large-scale collection of image-text pairs from open-source videos by expert surgeons.

Creating high-quality text-image pairs from endoscopic surgery videos presents significant challenges. Many of these videos contain background elements such as surgeon operations or procedural animations that are unsuitable for constructing our dataset. Additionally, issues like poor lighting and low image/audio quality further complicate the task. Moreover, some videos lack audio altogether or feature irrelevant content for endoscopic procedures.

To address the above issues, we meticulously designed the data collection and cleaning pipeline to ensure accurate image-text correspondence. Additionally, during the caption cleaning and question-answer construction, we collaborated with a team of expert clinical physicians to annotate and verify the accuracy and factual correctness of each pair, leveraging their domain expertise. Fig. 1 provides an overview of our dataset curation pipeline.

What sets our dataset apart is the combination of high-quality, large-scale video content and a meticulously designed data pipeline. In total, this dataset comprises 590 endoscopy videos, totaling more than 91 hours in duration, and 157,589 image-text pairs. Each image frame is accompanied by caption and question-answer correspondence designed from video captions, enhancing its utility for downstream surgical scene understanding tasks.

Another key aspect of our dataset is the detailed annotations provided by expert surgeons, which serve as invaluable repositories of human expertise in surgical procedure analysis and critical decision-making points. These captions encapsulate advanced information essential for artificial intelligence systems to not only comprehend, but also effectively replicate the nuanced decision-making processes inherent in surgical contexts. Through EndoAssistant dataset, we aim to catalyze advancements in multimodal learning and medical image understanding, empowering researchers to develop more robust and clinically applicable AI systems for endoscopic surgery.

In summary, our contributions can be outlined as follows: **i)** We meticulously curated a large-scale vision-language dataset (EndoAssistant) for endoscopic surgery comprehension, comprising 65,844 unique images, 30,002 unique captions and 157,589 image-text pairs in collaboration with expert surgeons, following rigorous data cleaning and optimization procedures. **ii)** We conduct extensive experiments using our proposed EndoAssistant (image-caption and image-question-answer pairs), demonstrating superior performance compared to previous methods on publicly available datasets. Our model utilizes contrastive learning to learn meaningful image-text corresponding representations for surgical scene understanding tasks, including cross-modal retrieval, image classification, and visual question answering.

## 2 RELATED WORK

**Vision-Language Learning.** Vision-language joint learning has been making significant strides in advanced multimodal understanding (Yin et al., 2023). A popular approach involves establishing connections between visual and textual information. Contrastive pre-training, exemplified by methods like (Wang et al., 2022a; Bao et al., 2022; Li et al., 2022; Radford et al., 2021), is one such technique. It works by simultaneously projecting text and image pairs into a shared embedding space, allowing for the discovery of relationships between the two modalities using cosine similarity. Another line of research focuses on Multimodal Large Language Models (MLLM) (Achiam et al., 2023; Team et al., 2023; Huang et al., 2023), which scale up generative pre-training (Brown et al., 2020) by incorporating both language and vision data. Early approaches like Flamingo (Alayrac et al., 2022), LLaVA (Liu et al., 2023), and MiniGPT-4 (Zhu et al., 2023) utilize pre-trained vision encoders and expand LLMs by incorporating visual tokens through visual instruction tuning. More recent advancements (Sun et al., 2023a; Lin et al., 2024; Yao et al., 2024; Wang et al., 2023; Dubey et al., 2024; Bai et al., 2023a) introduce more sophisticated joint training stages, significantly boosting performance. All these methods rely on high-quality vision-language datasets.

**Vision-Language Model and Datasets for Surgical Scene Understanding.** Surgical scene understanding is a complex task that involves multiple components: visual reasoning (i.e., identifying surgery phases and steps) (Seenivasan et al., 2022; Czempiel et al., 2021; Gao et al., 2021; Ding & Li, 2022; Zhang et al., 2022a), object localization (i.e., determining the position and count of organs/tools) (Qu et al., 2024; Huang et al., 2024; Jin et al., 2020; Rodrigues et al., 2022), and anomaly detection (i.e., spotting abnormalities that occur during surgery) (Jalal et al., 2023). A popular research direction is to create a unified semantic embedding that links images with their corresponding texts or captions (Twinanda et al., 2016; Zhang et al., 2022b; Vaswani et al., 2017; Yu et al., 2018).

Among all the multimodal tasks for understanding surgical scenes, the Surgical Visual Question Answering (Surgical VQA) task has been gaining increasing attention due to its interactive nature. Inspired by recent advancements in MLLM (Multimodal Large Language Models) mentioned above, several works have emerged that specifically target surgical applications. Notable methods include Surgical-VQLA (Bai et al., 2023b), which uses gated vision-language embeddings to fuse multimodal features, and Surgical-LVLM (Wang et al., 2024), which fine-tunes the MLLM Qwen-VL (Bai et al., 2023a).

However, surgical vision-language datasets are not as developed as the surgical MLLMs. The most commonly used datasets include EndoVis-18 (Allan et al., 2020), PSI-AVA (Valderrama et al., 2022a), and Cholec80 (Twinanda et al., 2016). There remains significant potential to further expand these datasets, especially in the number of categories, question diversity, and other aspects.

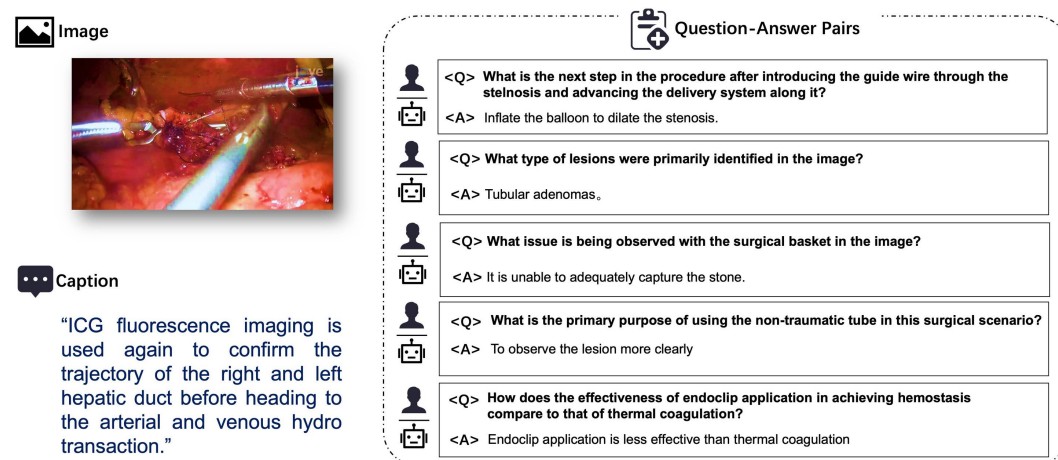

**Image**

**Caption**

"ICG fluorescence imaging is used again to confirm the trajectory of the right and left hepatic duct before heading to the arterial and venous hydro transaction."

**Question-Answer Pairs**

<Q> What is the next step in the procedure after introducing the guide wire through the stelnosis and advancing the delivery system along it?

<A> Inflate the balloon to dilate the stenosis.

<Q> What type of lesions were primarily identified in the image?

<A> Tubular adenomas。

<Q> What issue is being observed with the surgical basket in the image?

<A> It is unable to adequately capture the stone.

<Q> What is the primary purpose of using the non-traumatic tube in this surgical scenario?

<A> To observe the lesion more clearly

<Q> How does the effectiveness of endoclip application in achieving hemostasis compare to that of thermal coagulation?

<A> Endoclip application is less effective than thermal coagulation

Figure 2: Examples of the constructed EndoAssitant dataset. Each video frame contains its corresponding caption and question-answer pairs. We also design multiple choice questions upon the captions beyond the open questions showed above.

## 3 APPROACH

Creating a vision-language dataset from endoscopy videos is a complex task that requires careful design and refinement to extract high-quality, informative image-text correspondences from inherently noisy online sources. Publicly available endoscopy videos often lack audio and suffer from low image quality, making it difficult to extract scalable, expert-level data.

For instance, video frames containing endoscopic surgical images are frequently accompanied by irrelevant visual content, such as patients, surgeons, lecturers, surgical instruments, or procedural animations used for educational purposes. Text extraction poses an even greater challenge, as conventional automatic speech recognition (ASR) systems often struggle to meet the specialized demands of medical transcription. Both image and text de-noising add further complexity, requiring surgeon verification and automation tools to ensure both the quantity and quality of the dataset. As a result, using simple tools to extract frames at intervals is insufficient to capture the data accurately. As a result, using simple tools to extract frames at intervals is insufficient to capture the data accurately.

To collect the EndoAssistant dataset, we developed custom algorithms that leverage trained models to enable a semi-automated, surgeon-in-the-loop data curation pipeline, ensuring precise collection and alignment of both image and text modalities for downstream applications.

### 3.1 ENDOCAPTION: EXTRACTING IMAGE AND CAPTION PAIRS

Our data image-text extraction pipeline comprises several key steps: 1) Retrieving video sources covering the endoscopic surgery domain from Youtube [1] and JoVE [2]; 2) Filtering for videos with image level pertinency; 3) Extracting and cleaning text sentences; 4) Extracting key frames; and 5) Integration of both modalities.

**Collecting and filtering videos.** Using 'endoscopy' and 'endoscopic surgery' as retrieval keywords, we collect 3372 videos from publicly available platforms, including multiple channels from YouTube and medicine parts from JoVE. The diverse sources ensure a broad and representative dataset. First, we used the Silero Voice Activity Detection (VAD) model [3] and only consider the video with audios available. Second, we manually evaluated the overall video quality using metrics such as resolution,noisiness and frame rate. Videos with poor resolution or significant visual noise were excluded to avoid introducing low-quality frames into the dataset. To ensure the quality of relevant

---

[1] https://www.youtube.com/

[2] https://www.jove.com/

[3] https://github.com/snakers4/silero-vad

endoscopy content, we retained only those videos among all the downloaded candidates that had more than 40% of their frames classified as endoscopy images after our hierarchical image classification, which leads to a total of 690 videos after filtering.

**Hierarchical image classification.** We used pre-trained CLIP (Radford et al., 2021) `ViT-B/32` for coarse classification and further trained a fine classifier for endoscopic surgery images. For the training data, we leveraged 62k endoscopy surgery images from EndoSLAM (Ozyoruk et al., 2021), Endo-SfMLearner (Ozyoruk et al., 2021), and Kvasir (Pogorelov et al., 2017) as positive samples. Additionally, we manually annotated 56k images crawled from PubMed [4], among which 55k are negative and 1k are positive. Initialized with the pre-trained Inception-v3 Szegedy et al. (2016), our binary classifier is fine-tuned on the data mentioned above. This careful curation ensures high quality and minimal instances of blur or poor lighting, to guarantee that the collected data meet the stringent standards required for precise analysis and modeling in surgical contexts.

During inference, the image first goes through the pre-trained CLIP given prompts: "an endoscopic or surgical image", "a radiology image", "a presentation slide or lecturer talking" and "Doctors or patients during surgery." This meticulously engineered the classification labels the identify and differentiate endoscopic content from irrelevant content. The image with one classification probability larger than 0.25 is classified as an endoscopy or non-endoscopy image. Otherwise, the image is fed into the self-trained binary classifier. To evaluate this hierarchical image classification pipeline, we manually curated 50k randomly sampled from all the extracted videos and got a classification accuracy of 99.9% on this validation dataset.

**Collecting and cleaning captions.** The audio file extracted from each filtered video is fed into the Whisper-large-v3 model (Radford et al., 2022) to perform Automatic Speech Recognition (ASR) and transcribed into text sentences. Each sentence is treated as one chunk together with the starting and ending timestamp.

To only retain the informative text with medical relevancy and exclude irrelevant or extraneous speech, we further processed each text sentence using GPT-4 giving the below instruction.

- *You are an expert in medical terminology, specializing in analyzing and refining medical descriptions. Your task is to identify and retain only the most relevant medical information from the caption, excluding any unnecessary or irrelevant content. Correct any errors to ensure medical accuracy. Do not introduce new information or alter the original meaning. If there is no medical relevant information, please keep it empty.*

The GPT prompt design is a surgeon-in-the-loop practice where we iteratively incorporate feedback from a surgeon to validate the clinical relevance of the text. To validate the efficacy of the automatic text cleaning pipeline, we randomly sampled 5k from the nonempty returns and manually checked all of them were medical relevant: providing significant surgical insights (e.g., describing a procedure, identifying anatomical features, or highlighting observations).

**Extracting keyframes.** The filtered videos were partitioned into analogous chunks with identical start and end times as the audio chunks. Using FFmpeg [5], the keyframes were extracted from each of the video chunks. The threshold for the software, which determines the key frame based on the minimum amount of visible change in the scene, was selected after extensive experimentation. Each extracted image $I$ is accompanied by video index $v$ and the chunk index $c$ of the corresponding video, and then goes through a hierarchical image classifier and gets a binary label $y$ identifying if it is endoscopy-related or not.

**Combining both modalities for correspondence.** Finally, we have a list of image-caption pairs $[(v_1, c_1, T_1, \{I_1^i\}), ., (v_n, c_n, T_n, \{I_n^i\})]$, where $T$ represents the text sentence and $I_n^i$ is the of the $i$-th images in the $n$-th chunk. One caption may correspond to more than one different images after we apply visual content difference detection. We note that the image-text correspondence has been meticulously preserved throughout the entire process. While the image and text data are initially processed through independent cleaning branches, both retain their original timestamps, providing a strict foundation for dual-modality alignment. In the final step, these rigorous timestamps are used to align the cleaned image and text data, ensuring both topic relevance and precise correspondence. In

---

[4]https://pubmed.ncbi.nlm.nih.gov/
[5]https://www.ffmpeg.org/

addition, one surgeon manually checked 5k randomly sampled transcribed text are matched with the associated time-aligned images, preserving the context of surgical actions and discussions.

## 3.2 ENDOQA: CONSTRUCTING IMAGE AND QUESTION-ANSWER PAIRS

From the image-caption pairs mentioned above, we further construct question-answering data that might be able to facilitate clinical decision-making, enhance diagnostic precision, and beyond in a more direct way. We collaborate with surgeons to design prompts that guides GPT to generate QA pairs from each caption text. We identified two primary issues during QA curation: (1) The QA pairs are being used without captions, even though they are generated from caption text. (2) QA pairs should only be used when an image is provided, and they must focus exclusively on visual elements that can be inferred directly from the image. After several rounds of iterative fine-tuning and feedback, the prompt for question-answer generation is fixed and shown below:

- *You are an expert in endoscopic surgery. Based on the following image description, create high-quality Q&A pairs that address both the surgical procedure and medical knowledge. Please return high-quality Q&A, either open or multi-choice question together with the answer. The multi-choice question includes a question, 4 options, and 1 correct answer. Please give the correct answer and its detailed explanation.*
  *Caption: "*****"*
  *Q&A Guidelines:*
  *- Questions should relate to the surgical procedure shown in the caption.*
  *- Use proper medical terminology.*
  *- Cover the surgical relevant topics like actions, instruments, phases, steps, organs, objectives, and any potential complications.*
  *- Assume the caption is not available and the images must exist when asking and answering.*
  *- Provide detailed, medically accurate answers that would be useful for a medical professional.*
  *Generate one Q&A pair that satisfies the above requirements.*

Our questions vary widely, encompassing topics including surgical instruments, organs, surgical actions, analysis of a specific procedure and so on.

Fig. 2 provides qualitative examples of the curated EndoAssistant dataset: EndoCaption and EndoQA.

## 4 ENDOASSISTANT DATA OVERVIEW

The collected EndoAssistant, a vision-language endoscopic surgery dataset, consists of over 65K unique images, 40K unique captions, and 157K associated image-text, establishing it as the largest repository for endoscopic surgery vision-language tasks to date.

**Video Sources.** Our dataset boasts a total of 590 videos spanning over 91 hours, covering gastrointestinal endoscopy, ENT endoscopy, ophthalmic endoscopy, neuroendoscopy, and beyond. Spanning a diverse range of procedures, each video within our dataset varies in duration, typically ranging from 2 minutes to approximately 10 minutes. The number of medical relevant text sentences per video is 68 on average. This extensive temporal coverage not only ensures a comprehensive representation of surgical scenarios but also facilitates robust model training and evaluation across a broad spectrum of procedural complexities and durations.

**Data Statistics.** We curated 157K endoscopy surgical image-caption pairs, associated with image-QA pairs. The qualitative and quantitive analysis in Fig. 3 highlights a significant distinction between our EndoQA dataset and existing repositories of endoscopic surgery VQA datasets. Specifically: (1) our dataset demonstrates a more diverse semantic distribution, as evidenced by the word cloud and category analysis; (2) in terms of data source, our dataset is derived from 91 hours of video and 65,844 relevant frames, far surpassing existing counterparts; (3) the questions in our dataset are, on average, considerably longer. This length disparity is a strong indicator of the increased complexity inherent in our dataset, which not only requires a deeper understanding of surgical procedures but also demands more advanced reasoning capabilities from models tasked with answering these queries. Consequently, our dataset presents a more challenging and invaluable resource for advancing research in medical image understanding and multimodal reasoning.

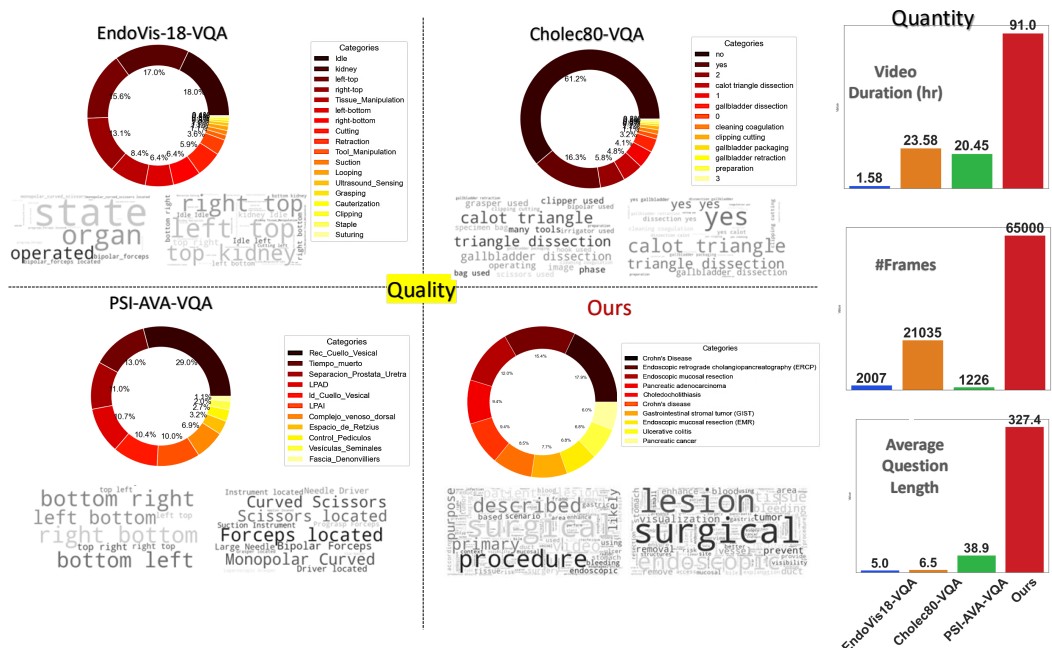

Figure 3: A comparison between the constructed EndoAssistant and other endoscopic surgery video datasets. Our EndoAssistant excels existing datasets in terms of quality assessment (**Left:** answer category distribution and knowledge diversity) and quantity assessment (**Right:** video duration, the number of extracted frames, and the average question length).

**Data Quality Samples.** In order to improve the quality of image-text pairs for downstream tasks, we have explored the potential opportunities to process the text modality using context-enriched sentence expansion. To be specific, we give a brief overview of text quality with the presence of examples:

- **Caption granularity.** Our endoscopic surgery dataset stands out for its inclusion of detailed captions, a distinctive feature absent in all other existing surgical video datasets. These captions are meticulously crafted by expert surgeons and encompass a wealth of knowledge, including surgical procedure analysis and background information integration. This expert knowledge serves as a crucial resource for models to emulate human expert decision-making processes effectively. To illustrate the granularity of these captions, we provide two caption examples:
  *Caption 1. The following contrast-enhanced low mechanical index endosynography shows a corresponding behavior of the microvessels in autoimmune pancreatitis as already displayed for the macro vessels by contrast-enhanced high mechanical index endosynography.*
  *Caption 2. Now remove the injection catheter away from the varix and the assistant begins to flush the needle continuously with distilled water.*

- **Question-answering granularity.** To design high quality questions for the endoscopic surgery VQA task, we design a semi-automatic QA construction enabling the expert physicians to leverage the robust question generation capabilities of LLM. This collaborative effort yields a diverse array of questions, spanning different types and aspects of medical scenarios, thereby enhancing their applicability and challenge levels in medical education. Specifically, our designed questions cover a wide variety of endoscopic surgery aspects, including surgical phases/steps, organs, surgical, actions, and the objective of surgeons. Below are some examples.
  *Surgical instruments: What is the primary advantage of using color Doppler sonography in surgical scenarios as demonstrated in the scenario where a feeding arterial vessel is observed within a tumor?*
  *Surgical actions: What is the most likely action a surgical team should undertake when encountering a completely different, life-threatening scenario during surgery?*
  *Surgical steps: During the procedure described, after ensuring that the snare is fully opened, what is the next step in managing the lesion according to the provided video frame caption?*

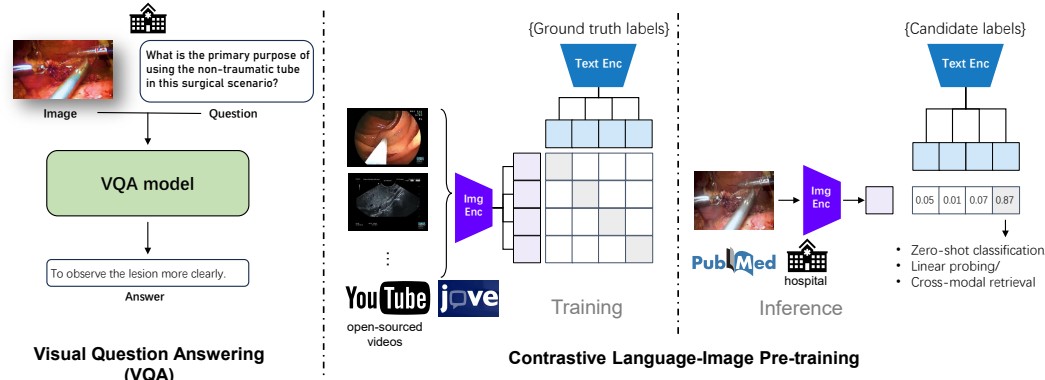

Figure 4: Illustration of the proposed **evaluation pipeline** for visual question answering (VQA) and contrastive language-image pre-training tasks. For both settings, we train the model with our curated datasets and test on publicly available datasets which are independent from our training set. This aims to demonstrate the expertise embedded in our dataset and its strong generalization ability.

***Surgeon objective:*** *In the context of this surgical scenario, what is the primary objective of targeting biopsies to small circular glands in contrast with foci of metaplasia?*

## 5 EXPERIMENT

To demonstrate the effectiveness and generalizability of our proposed EndoAssistant dataset, we develop both visual question answering (VQA) models (Sec. 5.2) and contrastive language–image pre-training (CLIP) models (Sec. 5.1) using our dataset. These models are externally tested on publicly available datasets, as illustrated in Fig. 4.

For the EndoCap dataset, we apply contrastive learning techniques, following the approach in Radford et al. (2021), to unify vision and language representations within a single embedding space for endoscopic surgery. We show efficacy of EndoCap dataset through the evaluation of various external domain applications, including cross-modal retrieval and both zero-shot and few-shot image classification.

In the case of the EndoQA dataset, we train a generalist foundation vision-instruction model, designed to handle diverse VQA tasks. This model facilitates the evaluation of out-of-domain endoscopic surgery VQA datasets, thereby providing insights into the model's ability to generalize beyond the specific domain of our training data.

### 5.1 CONTRASTIVE LANGUAGE–IMAGE PRE-TRAINING

In this subsection, we follow CLIP (Radford et al., 2021) to use our EndoCap dataset for contrastive language-image pre-training, where the training process simultaneously trains a text and an image encoder to increase the feature similarity from aligned pairs while decrease for misaligned pairs (Fig. 4 right).

We fine-tune CLIP with different model architectures on EndoCap and show that our data effectively enhances the image-language correspondence in the endoscopy surgery domain, consistently improving performance across multiple downstream tasks: cross-modal retrieval, zero-shot image classification, and few-shot image classification with linear probing.

**Baseline models.** We compare with MedCLIP (Wang et al., 2022b), PMC-CLIP (Lin et al., 2023b), BiomedCLIP (Zhang et al., 2024), CLIP (Radford et al., 2021), and SurgVLP Yuan et al. (2023; 2024a;b). Among the competing methods, MedCLIP is pre-trained on around 600K image-text pairs from the MIMIC-CXR (Johnson et al., 2019) dataset and CheXpert (Cherti et al., 2023) dataset, PMC-CLIP is pre-trained on around 642K data pairs from the PMC-OA (Lin et al., 2023b) dataset, BiomedCLIP is pre-trained on 15M data from the PMC-15M (Zhang et al., 2024) datasets, CLIP

(ViT-B/16) is pre-trained on the DataComp-1B (Gadre et al., 2023) dataset, and CLIP (ViT-B/32) is pre-trained on the LAION2B (Schuhmann et al., 2022) dataset.

**Evaluation datasets.** For cross-modal retrieval, we manually construct an endoscopy surgical image-caption data subset from two existing datasets: PathCap (Sun et al., 2023b) and PMC-CLIP (Lin et al., 2023b), of which the data are collected from PubMed and textbooks. For classification, four independent endoscopy surgical relevant anatomical detection datasets are used for external evaluation: Kvasir (Pogorelov et al., 2017), Hyper-Kvasir (Borgli et al., 2020), NBI-Inframes (Moccia et al., 2018), and GastroVision (Jha et al., 2023). Specifically, Kvasir contains 8K images from inside the gastrointestinal (GI) tract; Hyper-Kvasir contains 11K gastrointestinal images collected during real gastro and colonoscopy examinations; NBI-Inframes contains 720 images laryngoscopy from 18 different patients affected by laryngeal spinocellular carcinoma, and GastroVision contains 8K gastrointestinal images on colon, stomach, angiectasia, and esophagitis. The categories of Kvasir include polyps, two classes related to polyp removal and three anatomical landmarks in the GI tract. The Hyper-Kvasir dataset contains 23 classes including anatomical-landmarks such as cecum, ileum, pylorus, retroflex-rectum, retroflex-stomach and z-line. In the NBI-InFrames dataset, there are four classses include tissue with intraepithelial papillary capillary loop-like vessels, leukoplakia, hypertrophic vessels, and healthy tissue. GastroVision exhibits 22 classes, including anatomical landmarks, pathological findings, polyp removal cases and beyond.

Table 1: Cross-modal retrieval results on a cross-domain evaluation set consisting of 120 text-image pairs from PathCap (Sun et al., 2023b) and PMC-CLIP (Lin et al., 2023a) (endoscopy subsets verified by expert surgeons). Performances are measured by Recall@ (%). Bold and underlining highlight the best and second-best performances, while shading indicates models enhanced by our curated data.

| Methods | Text-to-Image | | | | | Image-to-Text | | | | |
|---|---|---|---|---|---|---|---|---|---|---|
| | R@1 | R@5 | R@10 | R@20 | R@50 | R@1 | R@5 | R@10 | R@20 | R@50 |
| MedCLIP (Wang et al., 2022b) | 0.83 | 4.17 | 9.17 | 16.67 | 42.50 | 0.83 | 5.00 | 7.50 | 16.67 | 29.17 |
| PMC-CLIP (Lin et al., 2023b) | 8.33 | 24.17 | 40.83 | 60.83 | **83.33** | **5.83** | 19.17 | 31.67 | 45.83 | 77.50 |
| BiomedCLIP (Zhang et al., 2024) | 2.50 | 8.33 | 10.83 | 19.17 | 43.33 | 0.00 | 5.00 | 8.33 | 20.00 | 42.50 |
| SurgVLP (Yuan et al., 2024b) | 0.83 | 5.00 | 15.83 | 23.33 | 52.50 | 0.83 | 5.83 | 10.83 | 23.00 | 58.33 |
| CLIP(ViT-B/32) (Radford et al., 2021) | 7.50 | 22.50 | 34.17 | 46.67 | 78.33 | 3.33 | 11.67 | 20.83 | 37.50 | 64.17 |
| CLIP(ViT-B/16) (Radford et al., 2021) | **9.17** | 26.67 | 35.83 | 53.33 | 78.33 | 2.50 | 16.67 | 26.67 | 37.50 | 68.33 |
| CLIP(ViT-B/32) + Ours | 4.17 | 20.83 | 38.33 | 53.33 | 78.33 | 4.17 | 16.67 | 35.00 | 49.17 | 75.00 |
| CLIP(ViT-B/16) + Ours | 7.50 | **33.33** | **63.33** | **75.83** | **83.33** | 3.33 | **25.00** | **45.00** | **62.50** | **87.50** |

**Implementation details.** For contrastive language-image pretraining, we employed a batch size as 512, a learning rate as 1e-5 with 200 warm up steps. The epoch for CLIP(ViT-B/32) and CLIP(ViT-B/16) is 19 and 14 respectively. For few-shot setting, we employed a batch size as 128, MultiStepLR learning scheduler with 0.1 as learning rate and 12 epochs. For linear probing, we employed a batch size as 128, CosineAnnealingLR scheduler using a learning rate of 0.02 and weigh decay as 5e-4 using 50 epochs.

**Cross-modal retrieval.** We compare the cross-modal retrieval results on a subset of 120 text-image pairs extracted from PathCap and PMC-CLIP (endoscopy subsets extracted by expert surgeons) as shown in Tab. 1. Noticeably, our fine-tuned model outperforms all competing methods over most recall thresholds. The superior performance indicates its ability to better capture and align cross-modal features, resulting in more accurate and relevant retrievals.

**Zero-shot image classification.** We compare the zero-shot image classificaiton results in Figure 5.a. Compared with the default CLIP pre-trained on natural images and other medical CLIP counterparts, our model fine-tuned with EndoCap significantly improves the classification accuracy, achieving the best results on the Kvasir, Hyper-Kvasir, and NBI-Inframes datasets.

**Linear probing for image classification.** We first use few-shot training samples with linear probing on NBI-Inframes datasets. As shown in Figure 5.b, our method consistently outperforms the counterpart methods when we increase the number of training data from 0 to 1, 4, 8, and 16. In addition, Tab. 2 compares the results leveraging all training samples with linear probing. Our fine-tuned model achieves the best results across all datasets, compared with different CLIP variants. Such performance advantages demonstrate the utility and generalization ability of our EndoCap dataset.

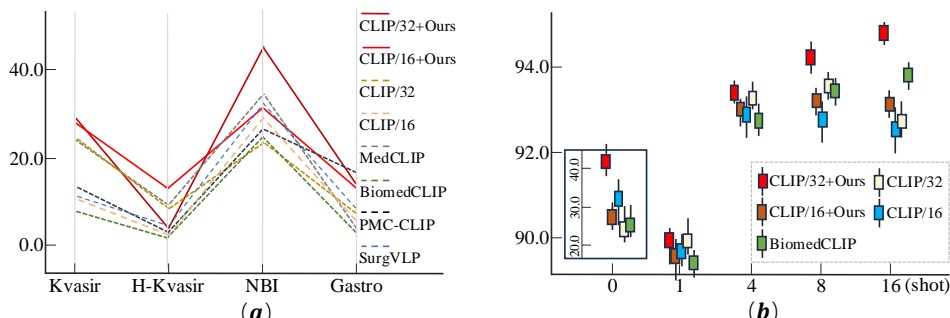

Figure 5: Comparisons of *N*-shot classification results. **(a):** A zero-shot classification performance (%) comparison across Kvasir Pogorelov et al. (2017), Hyper-Kvasir Borgli et al. (2020), NBI-Inframes Moccia et al. (2018), and GastroVision Jha et al. (2023). **(b):** The 0, 1, 4, 8, and 16-shot results on Fold1, Fold2, and Fold3 splits of the NBI-Inframes dataset Moccia et al. (2018). The box represents the mean performance (%), and the vertical line indicates the range of performance.

Table 2: Linear-probing results on Kvasir (Pogorelov et al., 2017), Hyper-Kvasir (Borgli et al., 2020), NBI-Inframes (Moccia et al., 2018), and GastroVision (Jha et al., 2023) datasets. Performances are measured by Accuracy (%). Bold and underlining highlight the best and second-best performances, while shading indicates models enhanced by our curated data.

| Model | Kvasir | Hyper-Kvasir | NBI-Inframes | | | GastroVision | |
| | | | Fold1 | Fold2 | Fold3 | Test | Val |
|---|---|---|---|---|---|---|---|
| PMC-CLIP Lin et al. (2023b) | 75.69 | 76.37 | 87.08 | 83.75 | 81.67 | 56.05 | 55.56 |
| BiomedCLIP Zhang et al. (2024) | 79.64 | 75.12 | 87.08 | 92.50 | 83.75 | 62.63 | 62.41 |
| SurgVLP Yuan et al. (2024b) | 72.24 | 72.93 | 92.5 | 92.5 | 88.33 | 59.15 | 59.30 |
| CLIP(ViT-B/32) + Ours | 83.29 | 85.51 | 95.42 | 95.83 | 90.83 | 71.44 | 71.30 |
| CLIP(ViT-B/16) + Ours | **85.25** | **86.40** | **97.08** | 95.83 | **93.33** | **72.13** | **72.06** |

## 5.2 Visual Question Answering

To evaluate the efficacy of the EndoVQA dataset, we conduct visual question answering (VQA) training and evaluate on two independent datasets: EndoVis18-VQA (Allan et al., 2020) and Cholec80-VQA (Twinanda et al., 2016), which focus on two types of question formats: open and binary classification (judging the correctness of statements). We measure the performances using accuracy, recall, and Marco-F1 scores between model outputs and ground truth.

We use LLaVA (Liu et al., 2024) as our baseline and present the results in Tab. 3. Starting with LLaVA-v1.5-7b, we finetuned with the default Lora finetuning hyper-parameters with 10 epochs. Notably, compared with the LLaVA model trained on EndoVis18-VQA and Cholec80-VQA, fine-tuning it on our EndoQA improves all three metrics, enhancing overall performance on both the EndoVis18-VQA and Cholec18-VQA datasets (Tab. 3). It is important to note that MedFuse (Sharma et al., 2021) and MFB (Yu et al., 2017) utilize models specifically designed for each dataset domain, whereas our approach employs a general-purpose vision-language model for all testing datasets. As a result, our model exhibits slightly lower accuracy and recall. However, despite this, our fine-tuned LLaVA achieves the highest F1 scores across both datasets. These results underscore the endoscopy expertise and broad generalization capabilities of our dataset.

## 6 Discussions

**Limitations.** Despite the promising results, EndoAssistant was curated using a combination of LLMs and handcrafted annotations. The automatic sections like the audio-to-text interpretation process and the subsequent text error correction can introduce biases and errors. Such errors are not rectifiable by the clinical experts in our current curation pipeline. Besides, the handcrafted sections can still benefit from automation. For example, developing a human-in-the-loop annotation pipeline using medical language models as a starting point could significantly accelerate the manual annotation

Table 3: Visual question answering results on EndoVis18-VQA (Allan et al., 2020) and Cholec80-VQA (Twinanda et al., 2016). Fine-tuning LLaVA on the proposed EndoAssistant improves the performance. Notably, specialized models tailor different models for each specific testing task, whereas our model is general-purpose: a single model for all tasks. Bold and underlining highlight the best and second-best performances.

| Type | Methods | EndoVis18-VQA(#2769) | | | Cholec80-VQA(#9096) | | | | Mean | |
|---|---|---|---|---|---|---|---|---|---|---|
| | | Acc | Recall | F1-score | Acc | Recall | F1-score | Acc | Recall | F1-score |
| Specialized | MedFuse (Sharma et al., 2021) | 0.6090 | 0.2610 | 0.2220 | 0.8610 | 0.3490 | 0.3090 | 0.8022 | 0.3285 | 0.2887 |
| | MFB (Yu et al., 2017) | 0.5238 | 0.4205 | 0.3622 | 0.8410 | 0.5303 | 0.4588 | 0.7669 | 0.5047 | 0.2887 |
| | SurgicalGPT (Seenivasan et al., 2023) | 0.6811 | 0.4649 | 0.4649 | 0.8746 | 0.5747 | 0.5794 | **0.8294** | 0.5491 | 0.5527 |
| General-purpose | LLaVA (Liu et al., 2024) | 0.5944 | 0.2988 | 0.2945 | 0.8457 | 0.5075 | 0.5001 | 0.7871 | 0.4588 | 0.4521 |
| | LLaVA (Liu et al., 2024) + Ours | 0.6475 | 0.3896 | 0.4008 | 0.8555 | **0.5991** | **0.5991** | 0.8069 | **0.5502** | **0.5528** |

process while ensuring professional accuracy at the same time. Another area for improvement lies in correlating caption text with temporal frame sequences rather than individual frames. Currently each caption is associated with an average of 5.25 frames. In this work we simply map each frame to the caption individually. However, adopting a sequence-to-text mapping could facilitate more precise learning of visual and textual embeddings. Given that existing benchmarks primarily focus on single image-text pairs, we aim to explore training with both sequence-text and image-text data in future work. One more area for improvement is to include data with challenging conditions often faced in endoscopic surgery, such as inconsistent lighting, obstructed or blooded views. During data collection, we focused on curating high-quality samples, filtering out cases with blurriness or poor lighting, which are indeed prevalent in real-world scenarios. Moving forward, we aim to incorporate more diverse and challenging cases into our dataset to enhance scalability, specifically targeting such challenging conditions.

**Conclusion.** In this work, we present an endoscopy-specialized large-scale vision-language benchmark (EndoAssist) assisting downstream tasks (cross-modal retrieval, image classification, VQA, *etc.*). The presented dataset is large-scale and diverse, sourced from open-source videos and verified by expert surgeons. Compared with existing endoscopic benchmarks, ours is more concentrated on surgical scenes in the operating rooms, in terms of question topics. The proposed image-caption pairs and image-question-answer pairs have demonstrated the superiority of our benchmark over mainstream vision-language pretraining frameworks through a broad range of empirical experiments.

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
