# OpenReview forum: "EndoAssistant: A Large-scale Vision-Language Dataset for Endoscopic Surgery Understanding from Open-Source Videos"
_ICLR.cc/2025/Conference — Submitted to ICLR 2025_

### Official Review · Reviewer_A9cJ · 2024-10-31

**Soundness:** 3
**Presentation:** 4
**Contribution:** 3
**Rating:** 6
**Confidence:** 4

**Summary:**

The paper introduces EndoAssistant, a large-scale vision-language dataset designed to enhance understanding of endoscopic surgery scenes. It addresses the limitations of existing datasets, which are small in scale and diversity, by providing a significantly larger collection of 590 videos, 65,844 unique images, 30,002 captions, and 157,589 image-caption/question-answer pairs. The dataset focuses on improving tasks like cross-modal retrieval, visual question answering (VQA), and image classification within the surgical context. The data curation process involves keyframe extraction, ASR transcription, hierarchical image classification, and rigorous text cleaning with clinical validation. EndoAssistant's vision-language data pipeline includes EndoCaption (image-caption pairs) and EndoQA (image-question-answer pairs), both of which are shown to improve baseline model performance across multiple benchmarks.

**Strengths:**

(1)EndoAssistant is the first large-scale, open-source vision-language dataset explicitly tailored for endoscopic surgery, surpassing previous datasets like Cholec80 in scale and semantic diversity. By integrating multiple existing models (CLIP, Whisper, GPT-4) into a surgeon-in-the-loop framework, this dataset provides a novel approach to generating diverse, medically relevant Q&A data from endoscopic videos.

(2)The paper clearly outlines each stage of the data pipeline, from video collection to model evaluation. The inclusion of figures detailing the dataset creation process and examples of the Q&A pairs and image captions adds clarity. Each stage is accompanied by performance metrics that demonstrate the impact of EndoAssistant on downstream tasks.

**Weaknesses:**

(1)Although EndoAssistant is curated for endoscopic tasks, some baseline models used (e.g., CLIP) are pre-trained on general vision-language datasets, which might limit their performance in highly specialized domains like medical imagery. Fine-tuning on similar medical datasets could make the evaluation more aligned with the dataset's intended use.

[1] Hecvl: Hierarchical video-language pretraining for zero-shot surgical phase recognition
[2] Procedure-Aware Surgical Video-language Pretraining with Hierarchical Knowledge Augmentation

(2) How does EndoAssistant perform on surgical tasks beyond classification and VQA, such as surgical phase recognition or anomaly detection?

(3) While the dataset draws from multiple open sources, there is a limited analysis of potential biases within the data. Different hospitals, surgical types, anatomical regions, or patient demographics could introduce significant variability, impacting the generalizability of the model.

(4) The dataset relies on relatively straightforward image-text pairing and may not fully capture deeper semantic alignment between the visual and language modalities (e.g., multi-level semantic alignment or co-occurrence patterns). Surgical procedures often involve subtle contextual changes, and certain tools or anatomical structures may carry different meanings across procedural stages.

**Questions:**

It is recommended to include in the limitations a discussion on the challenging conditions often faced in endoscopic surgery, such as inconsistent lighting, obstructed views, interference from bodily fluids, as well as data biases arising from differences in hospitals, types of surgery, anatomical regions, or patient demographics.

---

> ### Author Response · Authors · 2024-11-28
>
> > ### Fine-tuning on similar medical datasets could make the evaluation more aligned with the dataset's intended use. [1] Hecvl: Hierarchical video-language pretraining for zero-shot surgical phase recognition [2] Procedure-Aware Surgical Video-language Pretraining with Hierarchical Knowledge Augmentation
>
> Thanks for suggesting these related works. As the data collected in [1,2] are not released, we tested their released checkpoint (noted as surgVLP) under our settings and added comparisons in table1, table2, figure5.
> - Zero-shot cross-modal retrieval: text-to-image (T2I) and image-to-text (I2T)
>
> |             | **T2I**   |          |          |          |  |  | **I2T** |          |          |          |          |
> |---------------------------|---------------------|----------|------|----|----------|----------|----------|--------------------|----------|----------|----------|
> |                           | R@1    | R@5      | R@10     | R@20     | R@50   |  | R@1      | R@5               | R@10     | R@20     | R@50     |
> | SurgVLP | 0.83| 5.00     | 15.83    | 23.33  |   52.50   | | 0.83     | 5.83              | 10.83    | 23.00    | 58.33    |
> | CLIP (ViT-B/32) + Ours | 4.17| 20.83    | 38.33    | 53.33  |   78.33    || **4.17**     | 16.67             | 35.00    | 49.17    | 75.00    |
> | CLIP (ViT-B/16) + Ours | **7.50**| **33.33**    | **63.33**    | **75.83**   |  **83.33**|| 3.33| **25.00**         | **45.00**| **62.50**| **87.50**|
> - Zero-shot classification on four anatomical detection datasets
>
> |          | Kvasir | Hyper-Kvasir | NBI-Inframes | GastroVision |
> |--------------------------|--------|--------------|--------------|--------------|
> | SurgVLP                 | 10.95  | 4.80         | 31.11        | 6.97         |
> | CLIP(ViT-B/32) + Ours   | __27.51__  | 4.46         | __42.17__        | __12.35__        |
> | CLIP(ViT-B/16) + Ours   | 27.13  | __11.39__       | 28.89        | 10.26        |
> - Linear-probing on four anatomical detection datasets
>
> |                   | **Kvasir** | **Hyper-Kvasir** | **NBI-Inframes (fold1/2/3)**  | **GastroVision** |
> |--------------------------|--------|--------------|--------------|--------------|
> | SurgVLP | 72.24      | 72.93              |92.5/ 92.5/ 88.33         | 59.15  |
> | CLIP (ViT-B/32) + Ours  | 83.29      | 85.51             |95.42/95.83/90.83        | 71.44|
> | CLIP (ViT-B/16) + Ours  | **85.25**      | **86.40**             |**97.08/95.83/93.33**        | **72.13**|
>
> > ### How does EndoAssistant perform on surgical tasks beyond classification and VQA, such as surgical phase recognition or anomaly detection?
>  - Surgical phase recognition: EndoVis18-VQA and Cholec80-VQA datasets (table3) we used for visual question answering tasks contain **phase/tool recognition** questions.
>  - Kvasir, Hyper-Kvasir, NBI-InFrames, and GastroVision datasets (figure5 and table1,2) are all for **anatomical detection**. The categories of Kvasir include polyps, two classes related to polyp removal and three anatomical landmarks in the GI tract. The anatomical-landmarks in Hyper-Kvasir dataset include cecum, ileum, pylorus, retroflex-rectum, retroflex-stomach and z-line. The classes in the NBI-InFrames dataset include tissue with intraepithelial papillary capillary loop-like vessels, leukoplakia, hypertrophic vessels, and healthy tissue. GastroVision also exhibits a diverse range of classes, including anatomical landmarks, pathological findings, polyp removal cases and beyond.

---

> ### Author Response · Authors · 2024-11-28
>
> > ###  limited analysis of potential biases within the data, impacting the generalizability of the model.
> - Curated training data is unknown about the source hospitals and patient demographics, as such information is not available from video. They are all endoscopy-surgery: 91% Gastrointestinal Endoscopy, 6% ENT Endoscopy, 2% Ophthalmic Endoscopy, 1% Neuroendscosopy.
>  - To evaluate the generalizability, we evaluate on various testing data from different data origin and surgical types:
>
> |                           | **Kvasir**              | **Hyper-Kvasir**      | **NBI-Inframes**    | **GastroVision**      | **EndoVis18**      | **Cholec80**       |
> |---------------------------|-------------------------|-----------------------|---------------------|-----------------------|--------------------|--------------------|
> |Origin           | Norway                  | Norway                | Italy               | Norway, Sweden        |Germany                 | France             |
> |Types  | Gastrointestinal endoscopy | Gastrointestinal endoscopy | Laryngoscopy        | Gastrointestinal endoscopy | Colorectal endosocpy   | Laparoscopy   |
>
> > ###  The dataset relies on relatively straightforward image-text pairing and may not fully capture deeper semantic alignment between the visual and language modalities
>
> Our dataset generates question-answer pairs based on video captions, which contain deep semantic alignment between visual and language information. Compared to existing benchmarks that only contain simple question-answer pairs, our dataset introduces complex questions that require deep understanding of deep semantic alignment between visual and language information.
>
> > ###   include in the limitations a discussion on the challenging conditions
>
> Thank you for highlighting these challenges, which are critical for improving our work. During data collection, we focused on curating high-quality samples, filtering out cases with blurriness or poor lighting, which are indeed prevalent in real-world scenarios. Moving forward, we aim to incorporate more diverse and challenging cases into our dataset to enhance scalability, specifically targeting such challenging conditions. We believe that including these challenging scenarios will significantly improve the model's robustness in real-world applications. Regarding data bias and generalizability, we have addressed these aspects from both training and testing perspectives in our discussion. Additionally, we have included a detailed discussion of these common challenges in endoscopic surgery within the limitations section.

---

> > ### Comment · Reviewer_A9cJ · 2024-11-29
> >
> > Thank you to the authors for the clarification and the additional experiments, which addressed my concerns. As a result, I have raised my rating.

---

> > > ### Author Response · Authors · 2024-12-04
> > >
> > > Thank you once again! We sincerely appreciate your time and effort in reviewing and discussing our work.

---

> ### Comment · Reviewer_HEdX · 2024-11-29
>
> SurgVLP is a work on surgical phase and tool recognition. I do not understand why the authors compare it with zero/few-shot methods on the pathological and endoscopic diagnosis datasets. Comparing it with surgical understanding tasks is more reasonable.

---

### Official Review · Reviewer_VgBJ · 2024-11-01

**Soundness:** 2
**Presentation:** 3
**Contribution:** 3
**Rating:** 5
**Confidence:** 4

**Summary:**

The authors used a set of 590 endoscopic surgery videos to collect 157,589 image / caption pairs using a custom data curation pipeline. The image / caption pairs were further turned into open-ended and multiple choice question-answer pairs.

The utility of the two datasets (image / caption pairs and QA pairs) was validated by training a CLIP model and a LLaVa model respectively, and evaluating them on downstream tasks (zero-shot classification / retrieval and linear probe for CLIP and VQA for LLaVa), demonstrating comparable or superior performance than several other biomedical CLIP models and VQA models.

**Strengths:**

The authors' proposed dataset appears 5 - 10 fold larger than previous endoscopic surgery video datasets in terms of metrics like hours of video content sourced from, question length and number of frames.

The value of the dataset is validated across a range of different scenarios including zero-shot eval, representation learning (linear probe / few-shot learning) and VQA.

**Weaknesses:**

The quality of the dataset remains unclear to me - and would benefit from more clarification as well as investigation. While the dataset may be a valuable resource for computational researchers in the endoscopic surgical field, the paper otherwise does not appear to present novel ideas or evaluation. In fact, previous work published in CVPR 2024 have developed a much more sophisticated pipeline for curating instruction tuning data from Youtube videos in the field of pathology, involving more extensive quality control + mouse cursor location tracking: https://openaccess.thecvf.com/content/CVPR2024/papers/Seyfioglu_Quilt-LLaVA_Visual_Instruction_Tuning_by_Extracting_Localized_Narratives_from_Open-Source_CVPR_2024_paper.pdf.

For example, ASR is expected to be noisy and the transcript associated with a given key frame might have very limited context or be mismatched with the visual content displayed. Using GPT4 for retain only medically relevant information can help correct some incorrectly transcribed medical terms, but would not resolve the issues of limited context / mismatch with visual content. It is also not clear why out of 150k image / caption pairs, there are only 30,002 unique captions.

The created QA pairs similarly, might suffer from the same issues. I notice in the examples presented in Figure 2, are answers all very concise, and lack detailed explanation - which could arise due to both suboptimal prompting (e.g. only using zero-shot prompting instead of combining it with carefully, expert curated seed examples) and the concise nature of the source captions, and as a result limit their usefulness in training interactive AI assistants that can produce high quality responses in open-ended question answering (where a more detailed explanation or a specific response format is required).

Lastly, while the experiments are helpful for validating the usefulness of the dataset in the scenarios the authors investigated, crucial experimental details appear to be missing (i.e. hyperparemeters of training), which are needed to reproduce the results presented. Similarly, I cannot currently find a link to a github or hf repo that links to code and data used in the experiments in the study or the proposed dataset itself, and therefore can only draw conclusions about the quality of the data based on select examples / statistics presented in the paper.

**Questions:**

1. What is the average / std for the length of captions?
2. How are the 120 image / caption pairs used in "Cross-modal retrieval" selected? Will this data be made available for future works to allow comparisons?
3. Can the authors offer any additional insights about the quality of the dataset? (perhaps having experts review a random subset and rate accuracy / descriptiveness, etc.)
4. What is the motivation for using both a CLIP model and a custom pretrained CNN classifier to classify endoscopic vs. irrelevant content?

---

> ### Author Response · Authors · 2024-11-28
>
> > ###  What is the average / std for the length of captions?
>
> The average/std of the caption lengths is 15.7/11.7.
>
> > ###  How are the 120 image / caption pairs used in "Cross-modal retrieval" selected? Will this data be made available for future works to allow comparisons?
>
> Expert surgeons manually selected and verified an endoscopy subset from PathCap (Sun et al., 2023) and PMC-CLIP datasets (Lin et al., 2023). We will release the code, checkpoints, and the curated data (training and testing) to the public upon acceptance.
>
> > ###  Can the authors offer any additional insights about the quality of the dataset?
>   - Visual-Contextual Alignment: We aligned images and captions using video timestamps, providing a highly stringent and consistent alignment guarantee.
>   - Contextual Richness: The current image-caption pairs have achieved state-of-the-art (SOTA) performance across four anatomical detection datasets, excelling in all standard vision-language pre-training evaluation protocols: cross-modal retrieval (Table 1), linear probing (Table 2), and zero-shot/few-shot classification (Figure 5). This highlights their richness and effectiveness.
> - QA Conciseness: We observed some hallucinations when generating explanations alongside answers using GPT.
>   - Gathering clinicians' feedback for prompt design and professional evaluation/correction adds significant cost.
>   - The primary aim of this paper is to demonstrate the effectiveness of the collected data from video, minimizing additional input from GPT to better validate its utility.
>
> As a result, we chose to use only the data without explanations.
> - Expert Evaluation: On randomly sampled 5k data, one surgeon checked (1) cleaned text all containing medical relevant and high-informative information; (2) texts were matched with the associated time-aligned images to ensure alignment with surgical terminology and context; (3) automatically curated QA dataset achieves 100% surgical relevancy, 95% accuracy, and includes 62% of questions that require image reference to be answered.
>
> > ###  What is the motivation for using both a CLIP model and a custom pretrained CNN classifier to classify endoscopic vs. irrelevant content?
>
> The motivation for using a two-step filtering process involving both a general CLIP model and a custom specific classifier lies in their complementary strengths in handling diverse and domain-specific content respectively.
>
> Step 1: CLIP for Broad Filtering
> - Generalization Capability: CLIP is pre-trained on tons of data, effectively distinguishing broad categories like “surgical endoscopy image” vs. “radiology image” vs. “presentation slide or lecturer talking"  vs.  “Doctors or patients”.
> - Necessity to re-classify the images around boundary: given the aforementioned prompts, we noticed that the classification accuracy of pretrained CLIP is almost 100% when it gives a high confidence (probability > 0.25 for one certain class), otherwise it requires an additional classifier.
>
> Step 2: Custom classifier for Fine-grained Classification
> - Endoscopy and non-endoscopy training data: we collect positive samples from existing endoscopy datasets but the non-endoscopy images can be quite diverse and thus require self annotation. We manually labeled 56k images (enough to cover the diversity of the negative samples).
> - High Precision: We train a binary classifier specifically on the aforementioned data, which is able to refine identified subtle features unique to endoscopy, like organ textures and surgical lighting, removing residual non-endoscopic images missed by general CLIP.
>
> Overall, the introduction of CLIP alleviates our labor to annotate the training data for custom classifier. CLIP gives broad generalization capabilities for initial filtering and in-house annotated data gives domain-specific expertise for fine-grained classification. This ensures both scalability and accuracy in distinguishing endoscopic content from non-endoscopic content.
>
> > ###  Implementation details
> - Contrastive language-image pre-training: we have updated the paper including all the training hyper-parameters for pre-training, few-shot, and linear probing settings.
> - LLaVA: we used the default finetuning hyper-parameters provided by https://github.com/haotian-liu/LLaVA/blob/main/scripts/v1_5/finetune_task_lora.sh on llava-v1.5-7b with 10 epochs.

---

### Official Review · Reviewer_HEdX · 2024-11-02

**Soundness:** 3
**Presentation:** 3
**Contribution:** 2
**Rating:** 3
**Confidence:** 5

**Summary:**

The paper introduces EndoAssistant, a large-scale, expert-annotated vision-language dataset for endoscopic surgery, designed to enhance AI-driven medical training and surgical decision support systems.

**Strengths:**

1. The creation of a large-scale dataset with 65,844 unique images and over 157,589 image-caption pairs holds great potential to facilitate robust model training.

2. Incorporating both image-caption and image-question-answer pairs in the dataset supports diverse applications, from simple classification to complex question-answering.

3. The detailed data curation pipeline involving keyframe extraction, captioning, and alignment is methodologically sound for ensuring quality data preparation.

4. The involvement of domain experts throughout the data curation process helps ensure high factual correctness and relevancy.

**Weaknesses:**

1. The sampling method ignores temporal dynamics in endoscopic videos, potentially limiting the model's ability to perform cross-frame reasoning and handle dynamic scenes effectively.

2. The image-text sampling process does not fully capitalize on multimodal associations, possibly lowering the model's performance in complex surgical scene understanding that requires integrated visual-text analysis.

3. The paper adopts a lower-performing method from Surgical-VQA (MICCAI 2023 paper) without demonstrating previous SOTA performance, casting doubts on the model's comparative effectiveness in the surgical domain.

4. The paper lacks a discussion on the downstream task (surgery-specific tasks such as phase or tool recognition) performance of models trained on the proposed dataset, which are essential for assessing practical applicability in the surgical domain.

5. The criteria for assessing the quality of text and images in the dataset are unclear, which may raise questions about the reliability of the dataset.

**Questions:**

1. Endoscopic surgery videos contain a large number of dynamic scenes, but the sampling process mainly processes static images or single-frame data, without fully considering the temporal information of the video. The sampling method that lacks temporal association will cause the model to be insufficient in dealing with cross-frame reasoning, thereby limiting the performance of the model in dynamic scenes. Discussion/experiments may be added for this problem.

2. The sampling process segments and processes images and texts. Although contrastive learning is used to project images and texts into a shared embedding space, the sampling process does not fully utilize the multimodal associations between images and texts. In particular, the semantic associations between images and corresponding surgical step descriptions and tool instructions are not strengthened during sampling. This weakened multimodal association may limit the performance of the model in understanding complex surgical scenes, especially in tasks that require the integration of visual and textual information (such as complex scene question answering or context understanding). A detailed analysis could be made.

3. Based on Surgical-VQA (MICCAI 2023, first release EndoVis-18-VQA & Cholec-VQA dataset), the accuracy of EndoVis-18-VQA & Cholec-VQA has reached 0.632 & 0.898, respectively. However, this submission selects the method with the lowest performance in Table 1 of Surgical-VQA. They have not proved their SOTA performance on benchmark VQA datasets in the surgical domain. BTW, after the first release, the specialized models have reached even higher performance.

4. Surgical data science researchers may focus on some surgery-specific tasks, e.g., phase/tool recognition. The zero/few-shot performance of a VLM trained on the proposed dataset may be expected. Besides, fine-tuning LLaVA on different datasets but evaluating on the same benchmark surgical datasets can further demonstrate the effectiveness of the proposed dataset.

5. What are the criteria for confirming the quality of text and images? Are there definite criteria for filtering low-quality images? What criteria do doctors use for text annotation?

**Details Of Ethics Concerns:**

The dataset obtained from the website may have copyright problems.

---

> ### Author Response · Authors · 2024-11-28
>
> > ### The sampling method lacks temporal association
>
> It is indeed insightful to correlate caption text with temporal frame sequences rather than individual frames. During preprocessing, even after keyframe extraction (images with significant differences), each caption corresponds to an average of 5.25 frames.  Although adopting a sequence-to-text mapping would enable more precise learning of visual and textual embeddings, we simply map each frame to the caption and still achieve the SOTA performance across four anatomical detection datasets, excelling in all standard vision-language pre-training evaluation protocols: cross-modal retrieval (Table 1), linear probing (Table 2), and zero-shot/few-shot classification (Figure 5). We plan to explore sequence-text data in future work as existing benchmarks primarily evaluate single image-text pairs. Additionally, we have included a discussion of this limitation in the paper's discussion section. Thanks again for the insightful suggestion!
>
> > ###  Multimodal associations between images and texts.
>
> Apologies for the confusion; we did not perform any sampling. We would like to clarify that the image-text correspondence has been carefully maintained throughout the entire process. While the image and text data are initially processed through independent cleaning branches, both retain their original timestamps, which provide a strong foundation for dual-modality alignment. In the final step, these precise timestamps are used to align the cleaned image and text data, ensuring both topic relevance and accurate correspondence. Overall, we did not diminish any multimodal associations present in the raw video. On the contrary, the keyframe extraction process removes redundant and endoscopy-irrelevant frames, which enhances the multimodal association by eliminating noise.
>
> > ###  Add SurgGPT in comparison
>
> We have updated Table 3 to include SurgGPT (advanced version of surgicalVQA) in the comparison. Notably, specialized models like SurgGPT __tailor different models for each specific testing task__, whereas our model is general-purpose: __a single model for all tasks__. Despite the inherent disadvantage in such a comparison, our model still achieves superior result on Cholec-VQA and comparable result on EndoVis-18-VQA dataset.
>
> > ### Surgery-specific tasks
>  - EndoVis18-VQA and Cholec80-VQA datasets (table3) we used for visual question answering tasks contain __phase/tool recognition__ questions.
> - Kvasir, Hyper-Kvasir, NBI-InfFrames, and GastroVision datasets (figure5 and table2) are all for __anatomical detection__. The categories of Kvasir include polyps, two classes related to polyp removal and three anatomical landmarks in the GI tract. The anatomical-landmarks in Hyper-Kvasir dataset include cecum, ileum, pylorus, retroflex-rectum, retroflex-stomach and z-line. The classes in the NBI-InFrames dataset include tissue with intraepithelial papillary capillary loop-like vessels, leukoplakia, hypertrophic vessels, and healthy tissue. GastroVision also exhibits a diverse range of classes, including anatomical landmarks, pathological findings, polyp removal cases and beyond.
>
> > ### The zero/few-shot performance of a VLM trained on the proposed dataset may be expected.
>
> Table1 and Figure5 shows performance comparison on various datasets in zero-shot and few-shot settings respectively. Our models outperform all the counterpart models on four anatomical detection datasets under both settings.
>
> > ### Fine-tuning LLaVA on different datasets but evaluating on the same benchmark surgical datasets
>
> Table3 shows comparisons on LLaVA fine tuned with different datasets and tested on the same benchmark surgical datasets: ‘LLaVA’ represents fine tuning with the corresponding training set, while ‘LLaVA + ours’ represents fine tuning with our data plus training sets.

---

> > ### Author Response · Authors · 2024-11-28
> >
> > > ###  What are the criteria for confirming the quality of text and images? Are there definite criteria for filtering low-quality images? What criteria do doctors use for text annotation?
> > - Criteria for image quality
> >   - Video Quality: The overall video quality was assessed manually using metrics such as resolution, noisiness and frame rate. Videos with poor resolution, or significant visual noise like blur or poor lighting were excluded to avoid introducing low-quality frames into the dataset.
> >   - Relevance to Surgical Context: 1. We only retained those videos among all the downloaded candidates that had more than 40% of their frames classified as endoscopy images. 2. Only frames that visually capture meaningful surgical scenes were included during frame extraction.
> >   - We developed a hierarchy image classifier (L210-227) and evaluated the accuracy on self-curated 50K images (L227).
> > - Criteria for text annotation by medical professionals
> >
> > On randomly sampled 5k data, one surgeon checked (1) cleaned text all containing medical relevant and high-informative information; (2) texts were matched with the associated time-aligned images to ensure alignment with surgical terminology and context; (3) automatically curated QA dataset achieves 100% surgical relevancy and 95% statement accuracy.

---

> ### Comment · Reviewer_HEdX · 2024-11-29
>
> 1. Although models like MedFuse or SurgicalGPT are only trained and tested on one single test, they are also much more lightweight compared with MLLMs, with better deployment possibilities in narrow & accurate scenarios.
> 2. Zero/few-shot studies on surgical phase recognition are lacking (e.g., the authors may use AutoLaparo & MultiBypass140). The current surgical understanding evaluation has involved the related training set in the training process, which cannot demonstrate the power of MLLMs.
> 3. SurgVLP is a work on surgical phase and tool recognition. I do not understand why the authors compare it with the pathological and endoscopic diagnosis datasets. Comparing it with surgical understanding tasks is more reasonable.

---

> ### Author Response · Authors · 2024-12-04
> **Thank you for your insightful suggestions! We have incorporated additional experiments and sincerely appreciate your efforts in reviewing and discussing our work.**
>
> > ###  Comparison with specialized models
>
> Thank you for your suggestion! For a fair comparison, we use one single model to test on two phase/tool recognition datasets for overall performance. While validation on external testing datasets is a critical and necessary practice in real-world medical applications, specialized models often struggle with cross-domain testing (as shown below).   Our model demonstrates superior performance on in-domain testing for Cholec80-VQA and comparable results for EndoVis18-VQA. Moreover, we achieve a __significant improvement on overall performance (generalizability)__, calculated as a weighted average based on the dataset sizes (#2769 for EndoVis18-VQA, #9096 for Cholec80-VQA).
>
> |             | EndoVis18   |          |           | Cholec80|     |     |Overall|    |          |
> |---------------------------|---------------------|----------|------|----|----------|----------|----|----------|----------|
> |                           | Accuracy    | Recall      | F1     | Accuracy    | Recall      | F1  |Accuracy    | Recall      | F1 |
> |SurgGPT(EndoVis18)| 0.6811| 0.4649     | 0.4649    |0.0002   |0.0003     |0.5390    | 0.1591    |0.1087   |0.5217    |
> |SurgGPT(Cholec80)| 0.0722 |0.2057 |0.4489 |0.8746   |0.5747  |0.5794    | 0.6873    |0.4886    |0.5489 |
> |Ours|0.6475 | 0.3896   |0.4008     |0.8555   |0.5991     |0.5991    |**0.8069** |**0.5502** |**0.5528** |
>
> > ###  Zero/few-shot studies on surgical phase recognition
>
> As you suggested, we test accuracy on AutoLaparo and MultiBypass140 under zero-shot and few-shot (1/4/8/16) settings. Overall performance is calculated as a weighted average of the three metrics, using the number of images in each dataset as weights: AutoLaparo contains 28,060 images, while the Bern and Strasbourg subsets of MultiBypass140 include 92,842 and 139,324 images, respectively. Due to time constraints, we compare ours with the most competitive counterpart, SurgVLP (Yuan et al., NeurIPS 2024) for the few shot settings. __Notably, our model achieves the highest overall performance across all settings.__
>
>  - Zero-shot (%)
>
> |                   | **AutoLaparo** | **MultiBypass140-bern** | **MultiBypass140-strasbourg**  | **Overall** |
> |--------------------------|--------|--------------|--------------|--------------|
> | BioMedCLIP | 5.56      | **12.09**              |2.52         |8.73  |
> | SurgVLP | **23.57**      | 8.71              |12.18         |12.17  |
> | Ours (CLIP ViT-B/32)  |16.79     | 5.83             |6.17        |7.19|
> | Ours (CLIP ViT-B/16)   | 17.26      | 11.39             |**14.54**        | **13.71**|
>
> - One-shot
>
> |                   | **AutoLaparo** | **MultiBypass140-bern** | **MultiBypass140-strasbourg**  | **Overall** |
> |--------------------------|--------|--------------|--------------|--------------|
> | SurgVLP | **0.32**      | **0.32**            |0.26        |0.288  |
> | Ours  (CLIP ViT-B/32) |0.28     |0.26             |0.26        |0.262|
> | Ours  (CLIP ViT-B/16) | 0.30      | 0.28            |**0.37**        |**0.330** |
>
> - 4-shot
>
> |                   | **AutoLaparo** | **MultiBypass140-bern** | **MultiBypass140-strasbourg**  | **Overall** |
> |--------------------------|--------|--------------|--------------|--------------|
> | SurgVLP | **0.425**      | 0.38            |0.461       |0.428  |
> | Ours  (CLIP ViT-B/32)  |0.37     |0.408             |0.457        |0.430|
> | Ours  (CLIP ViT-B/16)  | **0.425**     | **0.445**            |**0.508**        |**0.477** |
>
> - 8-shot
>
> |                   | **AutoLaparo** | **MultiBypass140-bern** | **MultiBypass140-strasbourg**  | **Overall** |
> |--------------------------|--------|--------------|--------------|--------------|
> | SurgVLP | **0.47**      | 0.42            |0.50       |  0.468|
> | Ours  (CLIP ViT-B/32)  |0.41     |0.41             |0.48        |0.447|
> | Ours  (CLIP ViT-B/16) | 0.46    | **0.48**            |**0.54**        |**0.510**|
>
> - 16-shot
>
> |                   | **AutoLaparo** | **MultiBypass140-bern** | **MultiBypass140-strasbourg**  | **Overall** |
> |--------------------------|--------|--------------|--------------|--------------|
> | SurgVLP | **0.49**      | 0.44            |0.52       | 0.488 |
> | Ours  (CLIP ViT-B/32)   |0.44     |0.45             |0.51     |0.481|
> | Ours  (CLIP ViT-B/16)    | 0.48    | **0.51**            |**0.56**        |**0.534** |
>
> > ###  Anatomical detection tasks
>
> Anatomical detection is a critical subtask in surgical procedures: it serves as a foundation of computer-assisted surgery. Under one-shot and few-shot settings, __our model consistently demonstrates state-of-the-art performance across two surgical phase recognition datasets and four anatomical detection datasets__, highlighting its __remarkable generalizability__ for real-world surgical applications.

---

### Official Review · Reviewer_ivdQ · 2024-11-03

**Soundness:** 2
**Presentation:** 2
**Contribution:** 2
**Rating:** 6
**Confidence:** 5

**Summary:**

This paper presents the first large-scale, meticulously curated image-text dataset of surgical endoscopic scenes and demonstrates its effectiveness in downstream surgical endoscopic scene comprehension tasks.

**Strengths:**

1.	The curated dataset is a valuable resource for developing automated systems (e.g., LLMs, VLMs) to assist medical professionals in surgical endoscopic scenes.
2.	The paper employs a well-designed data processing pipeline, including rigorous data cleaning and optimization procedures, to generate high-quality image-text pairs from a large collection of endoscopic surgical videos.
3.	The CLIP model pretrained in EndoAssistant demonstrates superiority over mainstream vision-language pretraining frameworks through a broad range of empirical experiments.
4.	The paper is very clear and easy to follow.

**Weaknesses:**

1.	The proposed Visual Question Answering (VQA) models should be evaluated on internal datasets, such as parts of EndoAssistant, to better assess the endoscopic knowledge learned by the models. Evaluating solely on external datasets can only provide a limited view of the model's capabilities.
2.	The data, model, and training details should be openly released.

**Questions:**

1.	How are the Question-Answer Pairs constructed? Looking at the Question-Answer Pairs in Fig. 2, some pairs appear to be divergent questions unrelated to the images. Will such data (Question-Answer Pairs) improve performance on specific downstream tasks?
2.	The proposed Visual Question Answering (VQA) models seem more akin to a VLM model than a traditional VQA model. Could you clarify this distinction?
3.	All symbols used in the tables should be clearly defined, such as shading and underlining.

---

> ### Author Response · Authors · 2024-11-28
>
> > ### Evaluate on internal datasets.
>
> Thank you for the suggestion! It seems valuable to turn the surgeon-evaluated subset into a testing benchmark. However, for now, we have used all of it for training. We are considering setting aside some for testing once we collect new data and have it reviewed by the surgeon.
>
> > ### Open release.
>
> We will release the code, checkpoints, and curated data (including training data from open-source videos and testing image-caption pairs from PubMed) to the public upon acceptance.
>
> > ###  How are the Question-Answer Pairs constructed? Some QA not related to the images.
>
> As described in Section 3.2, we collaborated with surgeons to design prompts that guide GPT in generating QA pairs from each caption text. On a randomly sampled subset of 5,000 data points, a surgeon verified that the automatically curated QA dataset achieved 100% surgical relevance, 95% accuracy, and included 62% of questions requiring image references to be answered. We note QA pairs not directly related to the images also contribute to downstream tasks by embedding essential surgical knowledge, which is crucial for addressing related topics.
>
> > ###  The proposed Visual Question Answering (VQA) models seem more akin to a VLM model than a traditional VQA model.
>
> The model is built on a large multimodal foundation model (LMM) that supports multi-round conversations. The VQA task, involving a single round of conversation, represents a subset of LMM's capabilities. In our case, we fine-tuned and tested the model using a single round of QA, rather than the multi-round visual instruction tuning typically utilized in generative LMMs.
>
> > ###  All symbols used in the tables should be clearly defined, such as shading and underlining.
>
> Thanks for the suggestion! We have updated the paper accordingly.

---

> > ### Comment · Reviewer_ivdQ · 2024-11-29
> > **comment**
> >
> > Thank you for your response.
> >
> > For “How are the Question-Answer Pairs constructed? Some QA not related to the images,”
> > In VLM or VQA model, the question and content are related to the image in order for the model to perform cross-modal reasoning, combining information from both the image and text to generate an accurate answer based on the image content.
> >
> > “The proposed Visual Question Answering (VQA) models seem more akin to a VLM model than a traditional VQA model.”
> > The VQA task should also have the ability to engage in multi-turn conversation.
> >
> > Therefore, I am inclined to maintain my score, as the approach taken in the paper is not very innovative.

---

> > > ### Author Response · Authors · 2024-12-04
> > >
> > > > ### Some QA not related to the images
> > >
> > > We note that the QA content is entirely surgery-related and relevant to the images, though some questions can occasionally be answered without direct reference to the images. Including such data in the training set offers benefits:
> > >
> > > - Enhancing Single-Modality and Cross-Modal Capabilities: Improving text reasoning ability strengthens overall cross-modal reasoning, as many vision-language models (VLMs) build upon strong language models (LLMs).
> > >
> > > - Preserving Cross-Modal Alignment: Even in cases where the answer is directly derived from the question based on common medical knowledge, the QA content still aligns with the image context, ensuring meaningful cross-modal associations.
> > >
> > > > ### Multi-turn conversation.
> > >
> > > Apologies for the confusion. Our model is capable of engaging in multi-turn conversations. What we meant to say is that the existing endoscopy/surgery VQA benchmarks are designed for single-round interactions in our quantitative evaluation.
> > >
> > > > ### Innovation
> > >
> > > - __Large-scale dataset__: we create a first-ever image-caption dataset specifically for endoscopic scenes, and an image-question-answer dataset that offers greater size (157K pairs) and diversity compared to existing collections, which will be released to public upon acceptance.
> > >
> > > -  __First VL foundation model for endoscopy__: We developed a versatile endoscopy/surgery vision-language foundation model. Our model is evaluated through a comprehensive assessment, demonstrating state-of-the-art performance across four surgical phase/tool recognition datasets and four anatomical detection datasets.
> > >
> > > Thanks again! We appreciate your time for review and discussion!

---

### Meta-Review · Area_Chair_A9rf · 2024-12-18

**Metareview:**

This paper mainly presents a dataset from surgical endoscopic videos for medical scene understanding. Overall, the paper can be followed. However, there are many aspects that shall be improved. For example, we shall test the recent state-of-the-art algorithms on the proposed dataset and give a detailed discussion on how these methods perform and possible findings/new insights. This would help to readers and the researchers to understand how this dataset can be used in their studies, what shall be the setting, etc.

**Additional Comments On Reviewer Discussion:**

NA

---

### Decision · Program_Chairs · 2025-01-22

Reject